# SHAPER-PND trial: clinical effectiveness protocol of a community singing intervention for postnatal depression

Carolina Estevao [1], Rebecca Bind,[1] Daisy Fancourt,[2] Kristi Sawyer,[1] Paola Dazzan,[1] Nick Sevdalis,[3] Anthony Woods,[1] Nikki Crane,[4] Lavinia Rebecchini,[1] Katie Hazelgrove,[1] Manonmani Manoharan,[5] Alexandra Burton,[2] Hannah Dye,[6] Tim Osborn,[6] Lorna Greenwood,[6] Rachel E Davis [3], Tayana Soukup [3], Jorge Arias de la Torre,[3,7] Ioannis Bakolis,[3,8] Andy Healey,[9] Rosie Perkins,[10,11] Carmine Pariante[1]

**Correspondence to**
Dr Carolina Estevao;
carolina.estevao@kcl.ac.uk

## ABSTRACT

**Introduction** Postnatal depression (PND) affects approximately 13% of new mothers. Community-based activities are sought after by many mothers, especially mothers that prefer not to access pharmacological or psychological interventions. Singing has shown positive effects in maternal mood and mother–child bonding. The Scaling-Up Health-Arts Programmes: Implementation and Effectiveness Research-Postnatal Depression study will analyse the clinical and implementation effectiveness of 10-week singing sessions for PND in new mothers. This protocol paper will focus on the clinical effectiveness of this trial.

**Methods and analysis** A total of 400 mothers with PND (with a score of at least 10 on the Edinburgh Postnatal Depression Scale) and their babies will be recruited for this hybrid type II randomised controlled trial. The intervention group will attend 10 weekly singing sessions held at community venues or online, facilitated by the arts organisation, Breathe Arts Health Research (Breathe). A control group will be encouraged to attend non-singing sessions in the community or online for 10 weeks. A package of assessments will be collected from participants for clinical, mechanistic and implementation outcomes, at different stages of the trial. Clinical assessments will include questionnaires and interviews for demographics, mental health and social measures, together with biological samples for measurement of stress markers; the study visits are at baseline, week 6 (mid-trial) and week 10 (end of trial), with follow ups at weeks 20 and 36. Multiple imputation will be used to deal with possible missing data and multivariable models will be fitted to assess differences between groups in the outcomes of the study.

**Ethics and dissemination** Ethical approval has been granted by the London-West London and GTAC Research Ethics Committee, REC reference: 20/PR/0813.

**Trial registration number** NCT04834622; Pre-results.

## Strengths and limitations of this study

► Largest cohort of participants in a study examining singing in postnatal depression.
► Largest cohort studying biological measures in relation to an arts intervention for postnatal depression.
► Largest study to analyse the effect of an arts intervention on the mother–infant interaction.
► The state-of-the-art design will result in a unique package of clinical and implementation effectiveness on this intervention.
► There may be inconsistency throughout an intervention period if in-person sessions are switched online due to COVID-19 social distancing restrictions.

pharmacological treatment has yielded positive results, these are hampered by low uptake and adherence, and while psychotherapy has produced mixed results, poses similar challenges, including short-lived improvements.[2–5]

Prior studies have found that PND carries implications for not only maternal health and well-being, but also for the mother–infant relationship.[6] A common finding is that, while interacting with their babies, depressed mothers spend more time in negative, withdrawn states[7 8] and exhibit reduced sensitivity,[9 10] and in turn, their infants mirror this behaviour by disengaging and adopting a passive behavioural style.[7] Moreover, depressed mothers are more likely to report reduced feelings of attachment toward their infant[11] and are less likely to mentalise appropriately with their newborn.[12] Maternal sensitivity is highly predictive of infant secure attachment,[13] which has long been linked to optimal developmental outcomes including mental health;[14] thus, intervention early-on to ameliorate the mother–infant interaction

## INTRODUCTION
### Postnatal depression

Postnatal depression (PND) affects approximately 13% of new mothers, with symptoms including low mood, fatigue, anhedonia, insomnia and irritability.[1 2] Although

is warranted in order to protect offspring future development. However, while home-based interventions have been successful in improving PND, they show limited efficacy in helping the relationship itself.[15] Indeed, a recent combined psychoeducation and play intervention in a community setting was found to decrease dysfunctional parenting behaviours, but the intervention proved more effective in PND symptom reduction than in interaction improvement;[16] and a very small case study found that mother–infant painting sessions were successful in improving both the relationship and PND symptoms, though the sample size was only four dyads.[17]

Despite these challenges in finding optimal treatment options, which can both alleviate maternal symptoms of PND and strengthen the mother–infant relationship, many mothers continue to engage in community group activities, such as mother–infant play groups, as they are found to be effective at relaxing mothers, providing good sources of social interaction, decreasing the monotony of each day, and providing a sense of personal fulfilment for mothers.[18] Specifically, there is a growing body of evidence demonstrating the effects of community group singing on mental health[19 20] (see also the Melodies for Mums (M4M) programme, described below). Singing to newborns is practised around the world, and research has demonstrated valuable benefits such as improving mother–infant interaction and reducing infant distress.[21–23] Listening to music during pregnancy is also associated with greater maternal well-being and reduced symptoms of PND in the first 3 months postbirth, and, additionally, daily singing to babies is associated with fewer symptoms of PND and increased maternal well-being, self-esteem and perceived mother–infant bond.[24] Consequently, there is a strong theoretical background for why singing could support mothers with PND. It is thought that postnatally depressed women may have difficulty with adherence to dyadic interventions given the nature and challenge of their symptoms[25]; however, no prior studies have assessed whether a community singing-based intervention may be able to both improve the quality of the developing relationship, as well as present as a more engaging intervention that mothers are likelier to complete.

### Biological underpinning of singing sessions for PND

Three biological pathways are suggested to underlie the benefits of music engagement on mood: the glucocorticoid-modulated stress system, the inflammatory system and the oxytocin system, which is also associated with maternal attachment and bonding. Of these, the hypothalamic–pituitary–adrenal axis, of which the hormone cortisol is a main measure, is the most well studied. Studies in various populations have given evidence for a reduction in salivary cortisol after a singing session,[26–29] pointing to a consistent stress-reducing effect of singing. Interestingly, one of these studies also showed that singing sessions produced increases in the levels of cytokines such as IL-(interleukin-)17, IL-2, IL-4

and tumour necrosis factor (TNF)-α, which correlated with the observed reduction in cortisol.[27] The evidence surrounding the oxytocin system and music participation is more conflicted. Oxytocin is generally shown to increase as a result of singing,[30 31] or even listening to music.[32] However, more recent studies have demonstrated a decrease in salivary oxytocin concentrations in certain circumstances, for example, after choir singing, but not solo singing.[28] In addition, one study showed a decrease in oxytocin after singing sessions, which was associated with the increase in cytokines and reduction in cortisol, as described above.[27] This was proposed to be indicative of a general dampening of all stress-related signalling, in which oxytocin also plays a part, leading to an overall improved sense of well-being.

Studies investigating the interplay of these three biological systems in relation to musical engagement during the perinatal period are sparse. Indeed, only one study has investigated the biological effects of maternal singing groups,[33] in which pregnant women were randomised to either music intervention, singing intervention or control group. This study found a decrease in cortisol in both intervention groups, but that the effect was stronger in the singing group. Additionally, both musical intervention groups showed an increase in salivary oxytocin post-session. This is of particular interest given oxytocin's involvement in the mother–infant relationship,[34] which may be strengthened in cases of mother–child biobehavioural synchrony, in which maternal and infant hormone levels covary.[35]

### The 'M4M' intervention

M4M is an intervention that was developed and tested as part of a collaboration between the Royal College of Music, Imperial College London and University College London (UK). M4M was tested in a three-arm randomised controlled trial (RCT) involving 134 mothers with PND (with an Edinburgh Postnatal Depression Scale (EPDS) score ≥10 at baseline, indicating depression), and compared the efficacy of community singing with a comparison group (10 weeks of creative play classes for dyads) and a wait-list control group (10 weeks of care-as-usual). This pilot study found that mothers with moderate-to-severe symptoms of PND, defined as an EPDS score of ≥13, who participated in the programme with their baby, had a significantly faster improvement in symptoms than mothers in usual care.[36]

Following the pilot study, M4M was taken-on as a service by Breathe Arts Health Research (Breathe) across the boroughs of Lambeth and Southwark, in South London. However, there has since been no further research on the programme; thus, its effectiveness on a larger scale remains to be evaluated, as do the acceptability, appropriateness and feasibility of its delivery on a larger scale, and its potential to be adopted and sustained as a cost-effective, beneficial service for mothers within national health services. In addition, there is no evidence on the effects of M4M on mother–infant interaction, and a detailed

mechanistic investigation of the biological underpinning of any therapeutic effects is also missing. This paper will focus specifically on the protocol for a Wellcome-Trust-funded study aimed to deliver the clinical effectiveness and the biomarkers evaluation of M4M, the 'Scaling-Up Health-Arts Programmes: Implementation and Effectiveness Research-Postnatal Depression' (SHAPER-PND) study. Information on the implementation effectiveness of M4M is discussed in a separate detailed protocol.

## METHODS AND ANALYSIS
### Aims
1. To determine the clinical effectiveness of the intervention in a larger sample size than previous studies (n=400 women and their babies). This will allow us to ascertain whether the initial findings from the pilot study are replicable on a larger scale. To understand whether singing is an efficacious treatment for PND, including on mother–infant interaction, various psychological, social and qualitative measures will be collected from women and their infants before, during and after the intervention period.
2. To ascertain the biological mechanisms by which postnatal singing groups putatively improve maternal mood and mother–infant relationships. This will further the current understanding of the positive effects of the intervention and provide objectively measurable outcomes. Maternal and infant samples will be collected to measure cortisol and oxytocin, plus maternal inflammatory markers.

### Objectives and outcomes
To address our aims, we have the following objectives and outcomes. As this protocol paper is focusing only on the clinical effectiveness of the trial, the objectives and outcomes of the implementation component of the trial have been reported in a separate manuscript.[37]

### Primary objective
1. Does community singing improve symptoms of PND?

Participants will complete the EPDS prior to the intervention (baseline) to ascertain the severity of their PND. The EPDS will be administered again at various time points throughout and following the intervention period (weeks 6, 10, 20 and 36).

Outcome: The primary outcome of this study is to understand whether singing reduces the severity of symptoms of PND on the EPDS between baseline and week 10 (end of intervention), via a reduction in total EPDS score.

### Secondary objectives
1. To assess whether singing improves (changes) further aspects of mental health, including depression using the Hamilton Depression Rating Scale (HDRS). Time frame: Compare change between baseline and weeks 6, 10, 20 and 36.
2. To assess whether singing improves (changes) further aspects of mental health, including depression using the Beck Depression Inventory (BDI). Time frame: Compare change between baseline and weeks 6, 10, 20 and 36.
3. To assess whether singing improves (changes) further aspects of mental health, including stress using the Perceived Stress Scale (PSS). Time frame: Compare change between baseline and weeks 6, 10, 20 and 36.
4. To assess whether singing improves (changes) further aspects of mental health, including well-being using the Office for National Statistics Well-being Scale. Time Frame: Compare change between baseline and weeks 6, 10, 20 and 36.
5. To assess whether singing improves (changes) further aspects of mental health, including anxiety, using the State-Trait Anxiety Scale. Time frame: Compare change between baseline and weeks 6, 10, 20 and 36.
6. To ascertain whether singing improves the observed mother–infant interaction using the Crittenden CARE-Index. Time frame: Compare change between baseline and weeks 10 and 36.
7. To ascertain whether singing improves the perceived mother–infant relationship using the Maternal Postpartum Attachment Scale. Time frame: Compare change between baseline and weeks 6, 10, 20 and 36.
8. To ascertain whether singing improves the perceived mother–infant relationship using the Parent Reflective Functioning Questionnaire. Time frame: Compare change between baseline and weeks 6, 10, 20 and 36.
9. To ascertain whether singing improves social support and reduces loneliness using the University of California Los Angeles (UCLA) Loneliness Scale. Time frame: Compare change between baseline and weeks 6, 10, 20 and 36.
10. To ascertain whether singing improves social support and reduces loneliness using the Multidimensional Scale of Perceived Social Support Time frame: Compare change between baseline and weeks 6, 10, 20 and 36.
11. To identify whether there are biological mechanisms underpinning the psychological outcomes assessed using changes in measurements in stress hormones, including hair cortisol, diurnal cortisol and salivary cytokines. Time frame: Compare change between week 1 and weeks 6 and 10.
12. To identify whether there are biological mechanisms underpinning the psychological outcomes assessed using changes in measurements in salivary oxytocin. Time frame: Compare change between week 1 and weeks 6 and 10.
13. To identify how the singing sessions affect the lived experience of mothers with PND using focus groups. Time frame: Qualitative data collection at week 10 (end of intervention).
14. To explore the phenomenology of PND and how singing intersects with PND among women with

**Table 1** Clinical effectiveness data collection timeline

| Study objective | Type of measure | Measures | Time point collected | Type of data |
|---|---|---|---|---|
| 1 | Mental health | Edinburgh Postnatal Depression Scale | Baseline, weeks 1, 6, 10, 20 and 36 | Quantitative |
| 2 | Mental health | Hamilton Depression Rating Scale, Structured Clinical Interview for DSM-IV Disorders | Weeks 1 and 10 | Quantitative |
| | | Beck Depression Inventory, State Trait Anxiety Inventory, Perceived Stress Scale; Office for National Statistics Well-being Scale | Weeks 1, 6, 10, 20 and 36 | |
| 3 | Social | Maternal Postpartum Attachment Scale, Parental Reflective Functioning Questionnaire | Weeks 1, 6, 10, 20 and 36 | Quantitative |
| | | Crittenden Child-Adult Relationship)* | Baseline, weeks 10 and 36 | |
| 4 | Social | University of California Los Angeles (UCLA) Loneliness Scale, Short General Self-Efficacy Scale, Multidimensional Scale of Perceived Social Support | Weeks 1, 6, 10, 20 and 36 | Quantitative |
| 5 | Biological | Diurnal cortisol saliva samples* | Baseline and week 10 | Quantitative |
| | | Presession and postsession saliva samples* | Weeks 1, 6, 10 (at sessions) | |
| | | Hair samples | Week 10 | |
| 6 | Qualitative | Focus groups with intervention groups to capture feedback from sessions and experiences from mothers in the groups, focusing on the lived experience of PND | Week 10 | Qualitative |
| 7 | Qualitative | Semi-structured interviews with a sub sample of the intervention group of women self-reporting particular risk factors for PND (partially informed by demographic questionnaires) | Weeks 10–20, informed by information collected at baseline | Qualitative |

*Collected from mothers and infants.
PND, postnatal depression.

particular risk factors for PND (traumatic birth, adverse childhood experiences and social isolation/loneliness) using semistructured interviews. Time frame: Qualitative data collection at week 10 (end of intervention).

### Data collection plan
The study is anticipated to run from September 2021 to December 2023.

In order to address the objectives within the clinical effectiveness arm of the study, data will be collected according to the data collection timeline in table 1.

In addition to the measures collected above, we will also collect comprehensive information on maternal and infant sociodemographics and medical health, maternal history of childhood maltreatment and current maternal abuse, in order to explore the interaction between notable risk factors and the development of PND.

### Trial design
SHAPER is a multidisciplinary research programme funded by the Wellcome Trust that is hosted by King's College London (Sponsor) and University College London. The SHAPER-PND study described here is Hybrid Type II Effectiveness-Implementation RCT, where

equal focus is placed on the effectiveness of the intervention and the effectiveness of its implementation.[38]

### Study flow chart
An overview of the study timelines and procedures has been summarised in the flowchart below (figure 1).

### Setting
This is a multicentre trial that will be run in locations across London (primarily in the South London boroughs of Lewisham, Lambeth and Southwark), specifically in children's or community venues or online. These venues will be thoroughly safety and risk assessed in light of COVID-19 and adherence to any COVID-secure measures will be closely monitored by Breathe.

In order to enrol a sufficiently large sample of women, there will be 12 blocks of the 10-week singing programme in total: 10 intervention blocks will be for the experimental group, and 2 blocks will be for women in the wait-list control group that wish to take part in the singing sessions. The two wait-list blocks will be offered singing sessions after a woman has completed the first 10 weeks of the study and no data will be collected regarding these singing sessions, apart from the control follow-up data that this group is expected to provide.

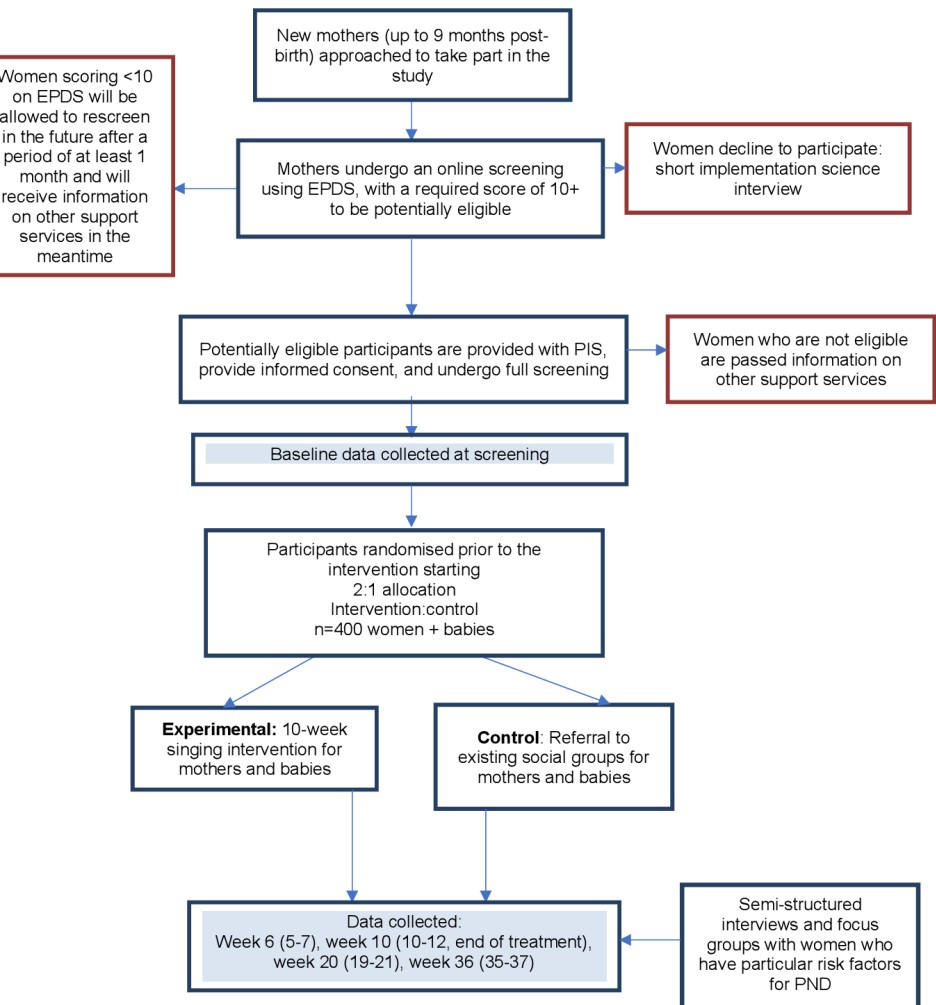

**Figure 1** SHAPER-PND study flow chart. EPDS, Edinburgh Postnatal Depression Scale; PIS, participant information sheet; SHAPER-PND, Scaling-Up Health-Arts Programmes: Implementation and Effectiveness Research-Postnatal Depression.

In the event that one or several women need to self-isolate due to confirmed or suspected exposure to COVID-19, the sessions will be delivered to her/them online via Zoom. This will allow for the sessions to still be conducted in person with the remaining women.

In the event of the artist or the Breathe officer having to self-isolate, a replacement Breathe-trained artist will be ready to step-in and deliver the sessions. The same will apply to Breathe staff; there will be another staff member available to be present at the sessions.

### Participants

Study participants will be new mothers exhibiting symptoms of PND, who meet criteria for a current major depressive episode, and have babies aged 0–9 months.

Inclusion criteria: Women aged 18 or older; satisfactory understanding of English; women with infants aged 0–9 months; women with PND, who score ≥10 on the EPDS. Women will require access to an internet-connected device (mobile phone, tablet, computer or laptop) to allow completion of assessments and participation in the singing sessions in the event they go online.

Exclusion criteria: The participant may not enter the study if her baby is outside of the age-range specified; if she is unable to give informed consent.

### Number of participants and power analysis

The pilot RCT that informed this study showed that depressive symptoms (EPDS scores) improved in both singing and control women by the end of treatment (week 10), but the singing group had a numerically larger improvement (−5.2 (SD=2.8)) than the care-as-usual group (−4.25 (SD=3.2), effect size=0.32)).[36] To have power of 80% (at p<0.05, two tailed) to detect the same effect size difference, randomising subjects 2:1, 232 subjects will be required in the intervention and 116 in the control group (total=348). In total, 400 mothers and their babies will be recruited to compensate for a 12% drop-out rate.

Participants will be recruited in blocks of 40–50 women and their babies, screened and assessed, prior to a 2:1 allocation randomisation (dyads will be allocated the same group). This process will be repeated 10 times, aiming to reach 400 dyads.

## Recruitment process

The recruitment period will last 2 years (2021–2022). All potential participants will initially be directed to a prescreening online form that includes an EPDS. If the EPDS score is <10, a participant will be notified that she is not currently eligible to participate and will be signposted to other support services within the community (eg, talking therapies, mother–baby groups, baby activity group). If a woman's score is ≥10, she will be notified that she is potentially eligible and will proceed straight to the second phase of screening through a home (or Zoom) visit. The participant information sheet for the trial will be sent once the baseline assessment is booked.

When a new round of 10 classes becomes available, researchers will arrange assessments with potential participants to undertake the full screening interview against the inclusion/exclusion criteria. If the participant is found non-eligible, she will be signposted to other support services within the community.

## The singing group

M4M is a 10-week intervention for mothers with PND. The programme will be delivered to groups of 8–12 mothers in weekly sessions lasting 1 hour. Mothers will start a block of 10-week classes together and continue with the same group and leader for the duration of the course. Classes are free to attend and will take place in Children's Centres or other community venues (or online via Zoom). Mothers will attend with their babies and will be invited to sit in a socially distanced circle on the floor surrounded by soft play cushions and mats. Classes will start with welcome songs, introducing the babies and mothers to one another, and then involve a range of singing and music activities. Mothers will be required to respect social distancing guidelines and keep their babies within their reach to avoid physical contact with other mothers or their babies.

Some songs will be accompanied by maracas, drums, hand chimes and other simple instruments that the mothers and babies can play together. Instruments such as guitar and ukulele will also be used for a small number of songs. Instruments will not be shared between the participants and will be disinfected before and after the singing sessions with a COVID-19 validated disinfectant.

Mothers will also work to write some of their own songs over the weeks, developing lyrics together about their babies or experiences of motherhood and creating simple melodies. Recordings of the group singing the songs together will be made and uploaded to private online platforms or onto CDs for the mothers to listen to at home. Classes will be led by professional workshop leaders trained by Breathe, with support of assistants.

## The control group

The control group will be an 'active' control. During the first 10 weeks (during the study period), mothers in the control group will receive details of other non-music classes available to them in the community (either in-person or online, depending on the programmes available at the time and government guidelines) and will receive the same schedule of texts and phone calls to encourage them to join these activities. They will still be seen by the researchers to collect clinical measures and biological samples (including the pre–post saliva samples) and to monitor engagement in other activities. Following the first 10 weeks, the mothers in the control group will be offered a place on the singing programme, but these data will not be part of the study, and they will not join groups with other women who are in the study.

## Patient and public involvement

Participants from previous sessions delivered by Breathe in the community have provided valuable feedback and insights into the sessions' delivery, content and schedule. These were gathered through focus groups and workshops where mothers and others involved in the singing sessions were able to share their experience of the sessions. This body of work allowed the refinement of the sessions for the intervention to be more suitable for the population.

A previous RCT was designed and delivered by two of this study's coauthors (DF and RP), along with Breathe.[36] The RCT analysed the clinical efficacy of the intervention on a smaller scale and explored the experiences of participants receiving and delivering the intervention, which ultimately informed the design of this SHAPER-PND study.

## Data analyses

### Quantitative analyses

The main statistical analyses will be carried out by the trial statistician who will be blind to the group randomisation until the main analyses are complete. The analyses outlined in this strategy will be primarily based on intention to treat. A per-protocol analysis will also be conducted in which the primary outcome will also be compared between groups, removing data of participants who:

1. Do not complete their intervention as defined above. 2. Miss over 50% of the singing or control sessions. 3. Are later found to have an exclusion criterion that was missed.

The first stage of analyses will be a descriptive model of the data to assess their completeness. Initially, a complete case analysis was performed under a missing at random assumption (MAR) where explanatory variables could predict the missing values in the respective outcome variable. These models drop any participant who does not provide outcome data at the follow-up time point.

Mother-level baseline variables will be described both overall and by randomised groups. Patterns of missing data will be described, if any. predictors of missingness will be also assessed. The primary outcome will be analysed using linear mixed models to model the change in total score of EPDS at 6 weeks, 10 weeks, 20 weeks and 36 weeks postrandomisation. A three-level hierarchical model will be employed with all-time points included as repeated measures in the model (6 weeks, 10 weeks, 20 weeks and 36 weeks postrandomisation) adjusting for baseline EPDS

to improve power and take into account clustering of the observation at mother's level. Associations between postrandomisation variables and missingness will be dealt with by multiple imputation by chained equations, again under the MAR assumption. Departures from this assumption will be assessed with a sensitivity analysis. All baseline characteristics as well as the outcomes will be included in the multiple imputation (MI) models. Secondary outcomes (eg, PSS, HAM-D) will be assessed with a similar methodology for the primary outcomes, using generalised linear mixed models depending on the type of outcome (normal, binary and count). The questionnaires to be used have validated methods of scoring, and the scores will be analysed as described. All analyses will be conducted in STATA V.15.1.

## Qualitative analyses

Data from the focus groups will be analysed using Thematic Analysis,[39] to understand the lived experience of PND and how this relates to experiences in the singing group. Data from the individual/small-group interviews with sub-groups of women self-reporting particular risk factors for PND (traumatic birth, adverse childhood experiences, and social isolation/loneliness) will be analysed using Interpretative Phenomenological Analysis[40] to focus in-depth on how singing intersects with the specific context of PND among women in these sub-groups.

## ETHICS AND DISSEMINATION

Ethical approval has been granted by the London-West London and GTAC Research Ethics Committee, REC reference: 20/PR/0813.

Samples will be handled according to HTA regulations. All data will be handled in line with GDPR and other relevant regulations.

Informed consent will be collected from all research participants and stakeholders involved in the study.

Findings will be published in peer-reviewed journals and disseminated at national and international meetings. Participants will be made aware of the results of the study through a newsletter.

**Author affiliations**
[1]Department of Psychological Medicine, King's College London Institute of Psychiatry Psychology and Neuroscience, London, UK
[2]Department of Behavioural Science and Health, University College London, London, UK
[3]Centre for Implementation Science, King's College London, London, UK
[4]Culture team, King's College London, London, UK
[5]South London and Maudsley NHS Foundation Trust, London, UK
[6]Breathe Arts Health Research, London, UK
[7]CIBER, Madrid, Comunidad de Madrid, Spain
[8]Department of Biostatistics and Health Informatics, King's College London, London, UK
[9]Health Service and Population Research Department, King's College London, London, UK
[10]Centre for Performance Science, Royal College of Music, London, UK
[11]Faculty of Medicine, Imperial College London, London, UK

**Correction notice** This article has been corrected since it was published. Lorna Greenwood has been added as an author and the contributors section has been updated.

**Contributors** All authors listed have contributed to the conception and design of the protocol and this manuscript. All authors have been involved in the drafting of the manuscript and have individually approved the version of the work published. Specifically, the contribution of each author falls within the following CRediT categories: CE, RB, KS, PD, NS, LR, KH, AB, RED, TS, JAdlT, IB and AH: conceptualisation, methodology and project administration. DF, RP and CP: conceptualisation, methodology, supervision and funding acquisition. NC and AW: conceptualisation, project administration and funding acquisition. MM, HD and TO: conceptualisation, project administration. LG: conceptualisation, project administration.

**Funding** This trial is part of the SHAPER programme, a Scaling-up Health-Arts Programme to scale up arts interventions. This programme is funded by the Wellcome Trust (award reference 219425/Z/19/Z). This work is additionally supported by the National Institute for Health Research (NIHR) Biomedical Research Centre at South London and Maudsley NHS Foundation Trust and King's College London and by a NIHR Senior Investigator to CP. NS, IB and AH' research is supported by the National Institute for Health Research (NIHR) Applied Research Collaboration (ARC) South London at King's College Hospital NHS Foundation Trust. NS and AH are members of King's Improvement Science, which offers cofunding to the NIHR ARC South London and comprises a specialist team of improvement scientists and senior researchers based at King's College London. Its work is funded by King's Health Partners (Guy's and St Thomas' NHS Foundation Trust, King's College Hospital NHS Foundation Trust, King's College London and South London and Maudsley NHS Foundation Trust), Guy's and St Thomas' Charity and the Maudsley Charity. RED is supported by the National Institute for Health Research (NIHR) Applied Research Collaboration: South London, at King's College Hospital NHS Foundation Trust.

**Disclaimer** The views expressed in this publication are those of the author(s) and not necessarily those of the NHS, NIHR or the Department of Health and Social Care.

**Competing interests** NS is the director of the London Safety and Training Solutions, which offers training in patient safety, implementation solutions and human factors to healthcare organisations. DF is a non-executive board director for Breathe Arts Health Research but she receives no financial compensation for her involvement. TS received funding for training cancer multidisciplinary teams in the assessment and quality improvement methods in the United Kingdom. TS serves as a consultant to F. Hoffmann-La Roche Diagnostics providing advisory research services in relation to innovations for cancer multidisciplinary teams and their care planning meetings in the United States of America. CP has received research and consultation funding from Boehringer Ingelheim and Johnson & Johnson for research on depression and inflammation, and by a Wellcome Trust strategy award to the Neuroimmunology of Mood Disorders and Alzheimer's Disease (NIMA) Consortium (104025), which is also funded by Janssen, GlaxoSmithKline, Lundbeck and Pfizer; the work presented in this paper is unrelated to this funding.

**Patient and public involvement** Patients and/or the public were involved in the design, or conduct, or reporting, or dissemination plans of this research. Refer to the Methods section for further details.

**Patient consent for publication** Not applicable.

**Provenance and peer review** Not commissioned; externally peer reviewed.

**ORCID iDs**
Carolina Estevao http://orcid.org/0000-0001-7758-0371
Rachel E Davis http://orcid.org/0000-0003-2406-7181
Tayana Soukup http://orcid.org/0000-0003-0203-7264

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
