## [Reviewer comments · BMJ Open]

ARTICLE DETAILS

TITLE (PROVISIONAL)	The SHAPER-PND trial: clinical effectiveness protocol of a community singing intervention for postnatal depression
AUTHORS	Estevao, Carolina; Bind, Rebecca; Fancourt, Daisy; Sawyer, Kristi; Dazzan, Paola; Sevdalis, Nick; Woods, Anthony; Crane, Nikki; Rebecchini, Lavinia; Hazelgrove, Katie; Manoharan, Manonmani; Burton, Alexandra; Dye, Hannah; Osborn, Tim; Davis, Rachel; Soukup, Tayana; Arias de la Torre, Jorge; Bakolis, Ioannis; Healey, Andy; Perkins, Rosie; Pariante, Carmine

VERSION 1 – REVIEW

REVIEWER	Dadi, A.F. Menzies School of Health Research, Centre for Child Development and Education
REVIEW RETURNED	31-May-2021

GENERAL COMMENTS	Thank you authors for planning this psychosocial components of intervention for reducing PND in new mothers. I am pleased to looking forward to the results of this trial. Saying this, I would raise my concerns that would affect the validity of the trial and a bit of clarity on few issues. 1. The abstract session does not clarify population for interventional group, who are they? I am sure these are mothers with PND and how this measured?2. No information about analysis plan3. I would delete the article summary section, specifically strength and and limitation, which are not fully known at this point.4. Primary objective, is a woman with an EPDS of 8 would be considered for the intervention? or is there any cut-off value for consideration? I would suggest following the standard cut-off values of the tool and include only women with high score to see the real effect of the intervention.5. In the secondary objective of the trial, what dimension of mental health would the BDI and HAM-D provide. How this is different from the primary objective using the EPDS?6. I am not sure how the investigators plan to handle variability of intervention in terms of modality (face-to-face Vs online), intensity(dose) of the intervention, different age of the women (two weeks to nine months)7. PND is considered for women from second week onwards to avoid mix-up of baby blue, severe psychiatric conditions so suggest to restrict the subjects to this age group
---

	8. The analysis part missed the type of analysis of RCT, intent to treat analysis or all randomized analysis? I am not sure how it is possible to assume missing is priorly determined.
--	---

REVIEWER	Devouche, Emmanuel University Paris Descartes, Psychology
REVIEW RETURNED	08-Jun-2021

GENERAL COMMENTS	Please clarify the recruitment process as I think something remains inconsistent between the text and the flowchart. The authors wrote that participants will be new mothers exhibiting symptoms of PND, who meet the criteria for a current major depressive episode (in addition to a score of ≥ 10 on the EPDS). I understand that all 400 women recruited are supposed to meet criteria i.e. women in the singing group and in the control group. In the flowchart, the N=400 new mothers are split in 'potentially eligible' (EPDS 10+) and EPDS <10; this seems contradictory to me with the inclusion criteria. Analysis of sociodemographic data is not specified, in particular the presence of a partner (married or not) and the socioeconomic level, both known as risk factors for PND. I also think very useful to consider ObGyn information (HTA, Diabetes, fetal distress, premature delivery threats, premature birth, ...) birth data (including physical abnormalities) if such data are available. Because the longitudinal follow-up will last 36 weeks, I also wonder whether authors planned to consider significant life events (either positive or negative) during the intervention period such (a new pregnancy, a move, a separation from the partner, a loss of a job, a death in close relatives...)? Additional data would also be interesting such as the risk for PND in partners but I guess it is too late to include it in the protocol. Research ethics are appropriately described in the abstract but the dedicated section is missing in the Method. This section needs thus to be reimplemented in the manuscript.
--

VERSION 1 – AUTHOR RESPONSE

Reviewer: 1

Dr. A.F. Dadi, Menzies School of Health Research, The university of Gondar, College of Medicine and Health Sciences Comments to the Author:

Thank you authors for planning this psychosocial components of intervention for reducing PND in new mothers. I am pleased to looking forward to the results of this trial. Saying this, I would raise my concerns that would affect the validity of the trial and a bit of clarity on few issues.

Reviewer 1 comment: The abstract session does not clarify population for interventional group, who are they? I am sure these are mothers with PND and how this measured?

Author response: We have clarified in the abstract that the population is mothers with PND, ascertained with a score of at least 10 on the Edinburgh Postnatal Depression Scale.

Reviewer 1 comment :No information about analysis plan.

Author response: The following paragraph was added to the methods and analysis section of the abstract: "Multiple Imputation will be used to deal with possible missing data and multivariable models will be fitted to assess differences between groups in the outcomes of the study." More detail is provided in the analysis section.

Reviewer 1 comment : I would delete the article summary section, specifically strength and and limitation, which are not fully known at this point.

Author response: We are grateful for your advice; however, this section has been kept as per the editor's instructions.

Reviewer 1 comment : Primary objective, is a woman with an EPDS of 8 would be considered for the intervention? or is there any cut-off value for consideration? I would suggest following the standard cut-off values of the tool and include only women with high score to see the real effect of the intervention.

Author response: In order to be eligible for our intervention, participants must score at least 10 on the EPDS. Previous studies have found that a score of 10 is sufficient to suggest postnatal depression, and furthermore our pilot study successfully used a cut-off score of 10 to identify depressed participants, who, by the end of the intervention, had a significant decrease in their EPDS score. We have listed this score in our inclusion criteria.

Reviewer 1 comment : In the secondary objective of the trial, what dimension of mental health would the BDI and HAM-D provide. How this is different from the primary objective using the EPDS?

Author response: The BDI and HAM-D will be used to collect additional dimensions of participants' depression that the EPDS does not quite establish. We are using the EPDS as our eligibility screening as it's specifically designed for postpartum women, and once participants are eligible, we will administer further depression interviews to collect as much information as possible about their specific symptom profiles.

Reviewer 1 comment : I am not sure how the investigators plan to handle variability of intervention in terms of modality (face-to-face vs online), intensity (dose) of the intervention, different age of the women (two weeks to nine months)?

Author response: If we must switch our sessions from in-person to online we will include this in our analyses so that we can see whether there are any differences in our outcomes among women who attended in-person sessions versus online. Regarding infant age, we will include that as a covariate in our analyses to explore whether infant age has played any role in our outcomes. Finally, we will include a variable in our analyses that indicates how many sessions each woman attended, which we can also use as a covariate if there is a wide range of sessions-attended.

Reviewer 1 comment : PND is considered for women from second week onwards to avoid mix-up of baby blue, severe psychiatric conditions so suggest restricting the subjects to this age group.

Author response: The Edinburgh Postnatal Depression Scale (EPDS) will be used as one of the screening criterium for inclusion in the study. This scale is specific for the postnatal period and focuses on the 7 days prior to questionnaire completion. We acknowledge that there is a risk that women suffering

from baby blues may be eligible according to this criterion, however baby blues typically goes away after a few days and thus is unlikely to skew a woman's questionnaire asking about an entire previous week. As a precaution, though, if a participant is recruited more than two weeks before the sessions begin we will repeat her EPDS to ensure that her depressive symptoms at the time of EPDS completion were not transient baby blues. Furthermore, if a woman meets the criteria for moderate-severe depression on the EPDS within the first two postpartum weeks, it is more likely that she is having a carry-over of antenatal depression than baby blues given that antenatal depression is a major risk factor for remaining unwell in the postpartum.

Reviewer 1 comment :The analysis part missed the type of analysis of RCT, intent to treat analysis or all randomized analysis? I am not sure how it is possible to assume missing is priorly determined.

Author response: The analysis plan has been described in more detail in the Data Analyses, specifically the qualitative analysis section has been reworded and expanded.

Reviewer: 2

Mrs. Emmanuel Devouche, University Paris Descartes Comments to the Author:

Reviewer 2 comment : Please clarify the recruitment process as I think something remains inconsistent between the text and the flowchart. The authors wrote that participants will be new mothers exhibiting symptoms of PND, who meet the criteria for a current major depressive episode (in addition to a score of ≥ 10 on the EPDS). I understand that all 400 women recruited are supposed to meet criteria i.e. women in the singing group and in the control group. In the flowchart, the N=400 new mothers are split in 'potentially eligible' (EPDS 10+) and EPDS <10; this seems contradictory to me with the inclusion criteria.

Author response: Thank you for your comment, we have corrected the inconsistency found in the flowchart, that could have led to a misinterpretation of our recruitment strategy. New Figure 1 has been uploaded demonstrating the changes made.

Reviewer 2 comment : Analysis of sociodemographic data is not specified, in particular the presence of a partner (married or not) and the socioeconomic level, both known as risk factors for PND. I also think very useful to consider ObGyn information (HTA, Diabetes, fetal distress, premature delivery threats, premature birth, ...) birth data (including physical abnormalities) if such data are available.

Author response: Sociodemographic data will be collected, including marital status, duration of the relationship (if applicable), level of education and income of both the mother and the partner. We will also collect medical history of mother and baby, health issues during pregnancy, information regarding delivery and use of health services (by mother and baby). We have not included this level of detail in the manuscript for any of the measures to be collected but the protocol registration (NCT) number is provided, and the protocol can be consulted on clinicaltrials.gov. However, we have included the following sentence in our manuscript:

“In addition to the measures collected above, we will also collect comprehensive information on maternal and infant socio-demographics and medical health, maternal history of childhood maltreatment, and current maternal abuse, in order to explore the interaction between notable risk factors and the development of PND.”

Reviewer 2 comment : Because the longitudinal follow-up will last 36 weeks, I also wonder whether authors planned to consider significant life events (either positive or negative) during the intervention period such (a new pregnancy, a move, a separation from the partner, a loss of a job, a death in close relatives...)?

Author response: In order to ascertain whether circumstances have contributed to or influenced our participants' mental health and outcomes, we will regularly be asking them to fill in demographic interviews, which collect comprehensive information for both mother and baby on sociodemographic circumstances, medical health, mental health, and significant life events. We will use this information to investigate possible confounders or additional predictors of our outcomes.

Reviewer 2 comment : Additional data would also be interesting such as the risk for PND in partners but I guess it is too late to include it in the protocol.

Author response: While we recognize that this would be a very poignant parameter to study in this context, the study has a very comprehensive battery of assessments, and we are cautious not to overburden the participants. We are considering a parallel pilot study to examine the mental health of partners as we agree that this is an under-studied topic.

Reviewer 2 comment : Research ethics are appropriately described in the abstract but the dedicated section is missing in the Method. This section needs thus to be reimplemented in the manuscript.

Author response: The section has now been added after the methods section, thank you for your comment on this.

VERSION 2 – REVIEW

REVIEWER	Dadi, A.F. Menzies School of Health Research, Centre for Child Development and Education
REVIEW RETURNED	18-Sep-2021
GENERAL COMMENTS	Thank you for addressing my comments, pleased to review this protocol and happy to hear the result.
REVIEWER	Devouche, Emmanuel University Paris Descartes, Psychology
REVIEW RETURNED	27-Sep-2021
GENERAL COMMENTS	Dear Editor The authors appropriately answered my comments. The authors planned a very interesting intervention and I wished to stress how important is their research as PND is a real public health concern. I look forward to reading the results. Best regards E. Devouche